# External Stimuli-Responsive Characteristics of Poly(*N,N′*-diethylacrylamide) Hydrogels: Effect of Double Network Structure

**DOI:** 10.3390/gels8090586

**Published:** 2022-09-15

**Authors:** Julie Šťastná, Vladislav Ivaniuzhenkov, Lenka Hanyková

**Affiliations:** Department of Macromolecular Physics, Faculty of Mathematics and Physics, Charles University, V Holešovičkách 2, 180 00 Prague 8, Czech Republic

**Keywords:** stimuli-responsive hydrogels, double network, poly(*N,N′*-diethylacrylamide), swelling, NMR spectroscopy, swelling/deswelling kinetics

## Abstract

Swelling experiments and NMR spectroscopy were combined to study effect of various stimuli on the behavior of hydrogels with a single- and double-network (DN) structure composed of poly(*N,N′*-diethylacrylamide) and polyacrylamide (PAAm). The sensitivity to stimuli in the DN hydrogel was found to be significantly affected by the introduction of the second component and the formation of the double network. The interpenetrating structure in the DN hydrogel causes the units of the component, which is insensitive to the given stimulus in the form of the single network (SN) hydrogel, to be partially formed as globular structures in DN hydrogel. Due to the hydrophilic PAAm groups, temperature- and salt-induced changes in the deswelling of the DN hydrogel are less intensive and gradual compared to those of the SN hydrogel. The swelling ratio of the DN hydrogel shows a significant decrease in the dependence on the acetone content in acetone–water mixtures. A certain portion of the solvent molecules bound in the globular structures was established from the measurements of the ^1^H NMR spin–spin relaxation times *T*_2_ for the studied DN hydrogel. The time-dependent deswelling and reswelling kinetics showed a two-step profile, corresponding to the solvent molecules being released and absorbed during two processes with different characteristic times.

## 1. Introduction

Stimuli-responsive polymer hydrogels that are able to respond to the changes in temperature [1], pH [2], humidity [3], light [4], specific ions or molecules [5], electrical fields [6], solvent [7] and ionic strength [8] have been extensively studied due to potential applications in the areas of drug delivery [9], microlenses [10], sensors [1] and artificial organs [11].

Among the various stimuli, temperature is the most studied in the field of stimuli-responsive polymers because of the key role of temperature in nature [12,13,14,15,16,17]. The thermosensitivity of polymer hydrogels is associated with a variable balance between different types of interactions, especially hydrogen bonds and hydrophobic interactions. At temperatures below the volume phase transition temperature, hydrogels absorb water to reach the swollen state and, above the transition temperature, they release water and shrink. On a molecular level, the volume phase transition (collapse) in crosslinked hydrogels is assumed to be a macroscopic manifestation of a coil–globule transition, as was shown for poly(*N*-isopropylacrylamide) (PNIPAAm) in water by light scattering [18]. Subsequently, temperature as a stimuli was found for other acrylamide-based hydrogels, such as poly(*N,N′*-diethylacrylamide) (PDEAAm) [19] or poly(*N*-isopropylmethacrylamide) [20].

The collapse of hydrogels can be induced not only by temperature but also by the composition of the solvent. The coexistence of two polymer phases differing in the conformation and concentration of chains in the swollen polymer network was predicted theoretically and experimentally proven on polyacrylamide (PAAm) networks swollen in acetone–water mixtures [7,21,22]. It was later found that the presence of charges on the PAAm chains plays a decisive role in the formation of a phase transition (collapse) of PAAm networks, leading to a jump-wise change not only in swelling, but also in mechanical properties [23]. The change in swelling properties with solvent composition was also found for PAAm hydrogels in water/alcohols mixtures [24]. Most thermosensitive hydrogels show a co-nonsolvency effect in mixtures of two good solvents, i.e., they are swollen in both pure solvents but shrink into a compact form in mixtures of these solvents. The co-nonsolvency phenomenon was studied in particular in PNIPAAm hydrogels in various water–organic solvent mixtures [25,26].

The conformational transitions of polymers can be induced by the presence of salts and their concentrations. A salt-induced phase transition in a hydrolyzed ionic PAAm hydrogel was detected in a water–organic solvent mixture [5]. The effect of the phase transition in the charged PNIPAAm in aqueous NaCl solutions on swelling and mechanical properties was described [27]. The influence of various salts was studied on the deswelling isotherms of thin films of photo-crosslinked PNIPAAm and PDEAAm [28].

It was reported that introducing a double network (DN) structure for various combinations of polymers is an effective approach to prepare stimuli-responsive hydrogels. Compared to conventional hydrogels or single network (SN) hydrogels, DN hydrogels are associated with improved mechanical properties as well as a high degree of swelling. DN hydrogels have an interpenetrating polymer network structure and the properties of these two networks, such as network density, rigidity, molecular weight and crosslinking density, exist in sharp contrast [29]. The enhanced mechanical properties of DN hydrogels are related to the asymmetric combination of two networks, when upon deformation, internal fractures in the first network are formed and act as additional crosslinkers [30].

DN stimuli-responsive hydrogels have been studied primarily with regard to their sensitivity to temperature and pH. DN PNIPAAm/PNIPAAm hydrogels containing inorganic polysiloxane nanoparticles [31] or comprising an ionized first network with electrostatic co-monomer [32] were investigated with regard to the influence of the hydrogel composition on the volume phase transition, morphology, equilibrium swelling, deswelling–reswelling kinetics and mechanical properties. A pH- and temperature-responsive DN hydrogel based on PNIPAAm and polyacrylic acid using graphene oxide as an additive were synthesized and the influence of additive and acid contents on various physical properties was studied [33,34]. Highly stretchable and tough thermo-responsive DN hydrogels composed of poly(vinyl alcohol)-borax and poly (AAM-co-NIPAAM) were characterized using FTIR spectroscopy and studied for their mechanical properties, thermal behavior, swelling property and thermo-responsive behavior [35]. Recently, we investigated the temperature response of DN hydrogels based on thermoresponsive PNIPAAm and PDEAAm [36,37]. This study showed that the temperature response of the studied DN hydrogels is tunable by the network crosslinking density.

In present work, we investigate SN and DN hydrogels composed of PDEAAm and PAAm with regard to sensitivity to various stimuli. Using swelling characteristics and NMR spectroscopy, the response to temperature, the presence and concentration of NaCl salt, and the composition of the water–acetone mixtures was studied. Macroscopic detection using swelling experiments was combined with NMR spectroscopy, which, especially in the case of two-component hydrogels, allows us to observe the behavior of each component separately on the molecular scale. Using NMR relaxation experiments, the different dynamic state of solvent molecules was detected. The swelling and deswelling kinetic experiments of DN hydrogel in various solutions were performed and the corresponding kinetic time parameters were determined.

## 2. Results and Discussion

### 2.1. Hydrogels Synthesis

The details of the preparation of SN and DN hydrogels were reported previously [37]. Briefly, first, the SN-D hydrogel was prepared by the redox polymerization of aqueous solutions containing a monomer, N,N′-diethylacrylamide (DEAAm) (c_DEAAm_ = 127.2 g∙L^−1^), a crosslinking agent, N,N′-methylenebisacrylamide (MBAAm) (c_MBAAm_ = 1.5 g∙L^−1^), an initiator, ammonium persulfate (APS) (c_APS_ = 1 g∙L^−1^), and a catalyst, N,N,N′,N′-tetramethylenediamine (TEMED) (c_TEMED_ = 15 g∙L^−1^). Afterwards, the DN-DA hydrogel was prepared from the specimen cut from SN-D hydrogels swollen to equilibrium in a large volume of aqueous solutions containing a second monomer, acrylamide (AAm) (c_AAm_ = 142.2 g∙L^−1^), a crosslinking agent (MBAAm) (c_MBAAm_ = 0.15 g∙L^−1^), a photoinitiator, and 2- oxoglutaric acid (OGA) (c_OGA_ = 0.15 g∙L^−1^) by UV irradiation between two glassy plates separated by a spacer of silicone rubber. After the synthesis, the hydrogel samples were thoroughly washed with a large amount of distilled water to remove residual unreacted reagents.

A schematic illustration of the preparation of the DN hydrogel PDEAAm/PAAm and the chemical structures of PDEAAm and PAAm is presented in Figure 1.

### 2.2. Effect of the Temperature on the Hydrogels’ Behavior

The temperature dependences of the swelling ratios of SN-D and DN-DA hydrogels are shown in Figure 1. Compared to the SN-D hydrogel, the transition region of DN-DA hydrogel is less steep, slightly shifted to higher temperatures, and the DN-DA hydrogel deswells less during temperature increases. The DN-DA hydrogel is thus less sensitive to temperature, which is caused by the hydrophilic chains of PAAm component. As we have shown in [37], a significant amount of water molecules interacting with PAAm units were detected in the collapsed DN-DA hydrogel structures at elevated temperatures, resulting in the high swelling ratio of DN PDEAAm/PAAm hydrogels.

Figure 2 shows the high-resolution ^1^H NMR spectra for the SN-D and DN-DA hydrogels detected under the same instrumental conditions at two temperatures (25 and 66 °C). The assignment of resonances to various proton types is shown directly in the spectra measured at 25 °C and it is the following: water signal (peak A), CH_2_ group of PDEAAm (peak B), backbone chain groups CH of PDEAAm and PAAm (peak C and C′, respectively), the backbone chain groups CH_2_ of PDEAAm and PAAm (peak D) and CH_3_ group of PDEAAm (peak E). The chemical structures of PDEAAm and PAAm with the assigned ^1^H NMR signals are shown in Figure 1.

At lower temperatures, the hydrogels are swollen in water and polymer chains are flexible and the NMR signals of all polymer units are clearly resolved. As it is seen from the ^1^H NMR spectra measured at a higher temperature, the signals B, C and E of the PDEAAm component are markedly reduced in their integrated intensities. Evidently, at elevated temperatures, the mobility of the PDEAAm units in rather compact globular structures is reduced and the corresponding NMR lines become too broad and undetectable in high-resolution NMR spectra [38,39]. On the other hand, the PAAm signals C′ and D in the spectra of the DN hydrogel (Figure 2b) change less with temperature.

Equation (3) was used to calculate the collapsed *p*-fraction of units with significantly reduced mobility. For *I*_0_, we used the values of the integrated intensities obtained at 25 °C and the correction for the fundamental decrease in the integrated intensity with increasing temperature as 1/*T* was included [37,39]. Figure 3 shows the temperature dependences of the *p*-fraction as obtained for the methylene CH_2_ signals in the PDEAAm units (signal B) and for the CH signals in the PAAm units (signal C′) in the SN-D and DN-DA hydrogels. It is seen that the maximum value of *p*-fraction *p*_max_ as detected for the polymer units of the temperature-sensitive PDEAAm in the SN-D and DN-DA hydrogels are equal to 1, which means that all PDEAAm units are immobilized and involved in collapsed globular structures. In comparison with the swelling experiments, the NMR-determined transition is much more steeper as these methods detect other processes during the temperature-induced transition in hydrogels. The time-consuming release of solvent molecules as detected by the swelling experiments results in a broad transition interval. Contrarily, NMR spectroscopy follows the relatively fast aggregation of polymer units, which leads to a sharp change in the dependence of the *p*-fraction on temperature [37].

The relatively high value of *p*_max_ = 1 as detected for the PDEAAm units of the DN-DA hydrogel could be a consequence of formation of heterogeneous structure in DN hydrogels as we have already reported for PNIPAm/PAAm and PDEAAm/PAAm hydrogels [36,37]. It has been shown that the agglomerates are formed during the formation of the DN structure, and the PDEAAm units in these structures are limited in their mobility and therefore do not contribute to high-resolution NMR spectra even at temperatures below the transition. The PDEAAm units, which remain mobile at lower temperatures, collapse upon subsequent heating and the integrated intensity corresponding to their NMR signals is thus reduced. Since NMR spectroscopy follows the change in the hydrated state mainly of these PDEAAm units, a relatively intense transition with the high maximum values of *p*-fraction is detected in contrast to a small temperature change in the swelling ratio (Figure 1).

Interestingly enough, it is evident from Figure 3 that *p*-fraction as detected for the CH signal of the PAAm units in the DN-DA hydrogel increases with temperature to the maximum value *p*_max_ ≅ 0.3. This means that approximately 30% of temperature-insensitive PAAm units are restricted in their mobility at higher temperatures. The increase in the *p*-fraction in temperature for the PAAm units is more gradual than for the PDEAAm units and it is probably connected with the process of water release, when thermodynamic and interaction conditions for conformational change occur for some PAAm units of the dehydrated hydrogel and these PAAm units gradually collapse into compact globular-like structures. Previously, we did not observe such behavior in IPN networks consisting of PDEAAm and PAAm, where the network density of the first and second components was very low and the NMR signals of PAAm practically did not change with temperature, which implies that virtually all AAm units showed high mobility even at elevated temperatures [40]. It is therefore possible that the densely crosslinked and temperature-sensitive PDEAAm component causes a part of the hydrophilic units of PAAm to pack into dehydrated structures, which leads to a limitation in their mobility.

### 2.3. Effect of Solvent Composition on the Hydrogels’ Behavior

In Figure 4, the swelling ratio of the SN-D and DN-DA hydrogels is shown as a function of the acetone content in water–acetone mixtures for the two temperatures of 25 and 45 °C. At room temperature, the swelling characteristics of the SN-D hydrogel composed of PDEAAm is not practically affected by the presence of acetone in mixture and the swelling ratio decreases slightly with increasing acetone content. At the higher temperature of 45 °C, a visible increase in the swelling ratio is observed (Figure 4a). For pure water and 20 vol.% acetone content, obviously the temperature effect will cause the collapse of the hydrogel and the swelling ratio thus shows low values. The increasing value of the swelling ratio with increasing acetone content is obviously related to the co-nonsolvency behavior of the PDEAAm hydrogels, where the addition of acetone to water will cause higher swelling ratios. According to our findings, the co-nonsolvency behavior of the PDEAAm hydrogels in a mixed water–acetone solvent has not been investigated, but the phase transition of PNIPAm polymer in water–acetone solutions was studied in [41]. Assuming that the PDEAAm polymer has a similar con-onsolvency behavior as the PNIPAm polymer, then apparently at higher temperatures and at the acetone content higher than 20 vol.%, the hydrogel enters a swollen phase.

In comparison with the SN-D hydrogel, the DN-DA hydrogel formed by the double network PDEAAm/PAAm shows a completely different dependence of the swelling ratio on the acetone content. Swelling ratios measured at both temperatures show a significant decrease in the region of 30–60 vol.% of acetone (Figure 4b). It is clear that the swelling behavior of the DN-DA hydrogel is mainly determined by the PAAm component, which is known to be sensitive to the water–acetone solvent composition [7,21]. As shown in Figure 4b, the DN-DA hydrogel has high values of equilibrium swelling in pure water. This behavior suggests that the attractive interactions between the polymer chain and the water molecules dominate over the attractive interactions between the polymer chains. In water–acetone mixtures, molecules of these solvents have an attractive interaction, leading to an increase in the free energy for polymer–polymer contact and induce the collapse of the polymer network [24].

Using Equation (3), we calculated the *p*-fraction for the PDEAAm and PAAm NMR signals dependent on the acetone content. For *I*_0_, we took values based on integrated intensities as obtained for the hydrogels in D_2_O solution at 25 °C. As it is shown in Figure 5, the SN-D spectra of the PDEAAm hydrogel did not change with increasing the content of acetone, leading to a *p*-value of 0–0.1 regardless of the water–acetone content. On the other hand, the PAAm signals in the DN-DA hydrogel show a significant increase in the *p*-fraction dependent on the acetone content; the *p*-fraction value varies from 0 for pure water to 1 for pure acetone. Both findings are consistent with the results of the swelling experiments (Figure 4). What is interesting and noteworthy is the third dependence in Figure 5 for the PDEAAm component signals in the hydrogel DN-DA. PDEAAm is insensitive to the composition of the water–acetone content when in the SN-D hydrogel (empty blue squares), but in the DN-DA hydrogel, the *p*-fraction increases from *p*_max_ = 0 for pure water to *p*_max_ ≅ 0.5 for pure acetone (full blue squares). The increase in *p*-fraction for the PDEAAm component is delayed when compared to the PAAm component, starting with up to 60 vol.% acetone. This behavior is similar to that found for the temperature dependence; the units of the component that is insensitive to the stimulus (if it is in the single hydrogel version) partially packed into immobile structures, but need a stronger stimulus to do so.

### 2.4. Effect of Salt on the Hydrogels’ Behavior

The equilibrium swelling ratios of the SN-D and DN-DA hydrogels as a function of NaCl salt concentrations are shown in Figure 6. It has been noticed that the swelling ratio of the SN-D hydrogel at both temperatures in different NaCl solutions decreased strongly with the increase in salt concentration from 0.01 to 0.05 M and reaches very low values ≅0.2 for salt concentration 3 M and higher (Figure 6a). This behavior is consistent with the previously established effect of NaCl on the deswelling of the PDEAAm hydrogels [28]. At a higher temperature, the swelling ratio is more suppressed due to the combination of the two stimuli to which the PDEAAm hydrogel is sensitive.

As previously found, nonionic PAAm hydrogels induce a far weaker salt-induced swelling change compared to the ionic PAAm hydrogels [5]. As it is seen in Figure 6b, the introduction of PAAm as the second component in the DN-DA hydrogel leads to a very gradual decrease in swelling to a value ≅5. At the lower temperature of 25 °C and a low salt content, there is a somewhat sharper decrease in swelling and it is probably mainly the PDEAAm units that are affected by salt. For salt concentrations higher than 1.5 M, the swelling drop is slower and the PAAm units are probably gradually affected, while no effect of the temperature is observed.

Similar to the dependence of the NMR spectra on the temperature and acetone content, we determined the *p*-fraction for different molar concentrations of NaCl using Equation (3) and, for *I*_0_, we took values based on integrated intensities as obtained for the hydrogels in D_2_O solution at 25 °C. Figure 7 shows the dependence of the *p*-fraction on the NaCl concentration as determined for the SN-D and DN-DA hydrogels at 25 °C. For the SN-D hydrogel, the *p*-fraction grows very fast in the NaCl concentration region 0–2 M and it reaches a maximum value *p_max_* = 1, which means that all PDEAAm units are packed in collapsed structures. The PDEAAm component in the hydrogel DN-DA shows a similar increase in the *p*-fraction and reaches a maximum value for the NaCl concentration of 3 M. Both findings fully correspond to the results obtained from the swelling experiments (Figure 5). The increase in the *p*-factor, and thus the decrease in the intensity of the high-resolution NMR signals of the PAAm groups in the DN-DA hydrogel, are much slower in the dependence of the NaCl concentration and the maximum value of the *p*-fraction only reaches a value of 0.5. This means that roughly 50% of the PAAm units in the double network are unaffected by NaCl and prefer to interact with water molecules. Therefore, the swelling ratio of the DN-DA hydrogel for the maximum concentration of NaCl reaches a relatively high value (Figure 6b).

### 2.5. Effect of All Stimuli on the Hydrogels’ Behavior

Figure 8 displays the swelling ratios that were detected for the SN-D and DN-DA hydrogels in different environments, i.e., in pure water and acetone, and for a maximum concentration of NaCl = 6 M at two different temperatures. The influence of various stimuli on the swelling behavior of hydrogels is thus clearly demonstrated. The SN-D hydrogel composed of PDEAAm shows temperature and salt responsiveness. Compared to the hydrogel swollen in pure water at room temperature, the swelling ratio in the salt solution decreases almost 10 times, and this change is much more pronounced than when the temperature is increased. This may be related to the earlier finding that the conformation of PDEAAm is more compact in the presence of NaCl than that in the presence of a salt-free solution [42].

The sensitivity to stimuli in the DN-DA hydrogel is significantly affected by the introduction of the second component and the formation of the double network. Due to the hydrophilic PAAm groups, the extent of deswelling after temperature increase is reduced. On the other hand, the DN-DA hydrogel has a 15 times smaller swelling ratio in pure acetone compared to water. This is certainly caused mainly by the presence of acetone-sensitive PAAm units, but due to the interpenetrating structure of the DN network, 50% of acetone-insensitive PDEAAm units also contribute to the collapsed structures (Figure 5), which may affect the deswelling extent of the DN-DA hydrogel in acetone.

### 2.6. NMR Relaxation of the Solvent Molecules

NMR relaxation experiments on the nuclei of solvents should generally provide information on the mobility of the solvent molecules, and consequently on polymer–solvent interactions. The dynamical behavior of the solvent molecules was studied using the measurements of the ^1^H NMR spin–spin relaxation time *T*_2_ on water (HDO) and acetone signals and the *T*_2_ values as obtained for the DN-DA hydrogel at various solutions are summarized in Table 1. Single-exponential relaxation decay characterized by single relaxation *T*_2_ was detected for the DN-DA hydrogel at the temperature of 25 °C (Appendix A). Heating at 45 °C leads to bi-exponential relaxation decay (Appendix A) and the *T*_2_ components of bi-exponential dependences differ significantly from each other and they can be marked as relaxation times of free (*T*_2_ = 5.0 s) and bound (*T*_2_ = 1.0 s) water. The main reason for these differences is that the motion of bound water in collapsed structures is spatially restricted and anisotropic [39], while free water molecules are either contained in less swollen polymer structures or released from the interior of the hydrogel. The occurrence of bound water molecules with slow motion was also previously detected in collapsed poly(vinyl methyl ether) and PNIPAm hydrogels [43,44,45,46,47] and interpenetrating PNIPAm-based hydrogels [36].

Single- and bi-exponential relaxation decays were also detected for water and acetone signals in water–acetone mixtures with the DN-DA hydrogel (Appendix A). For the 20 vol.% acetone solution, the solvent molecules show a single, relatively high relaxation time *T*_2_ value, which corresponds to their high mobility in the swollen structures of the DN-DA hydrogel. In 80 vol.% acetone solution, the DN-DA hydrogel is collapsed with a very-low swelling ratio (Figure 4b) and the solvent molecules are either in a free or bound state and, at the same time, the value of the relaxation time of bound molecules is up to three orders of magnitude lower than the relaxation time of the free molecules. This signifies that the collapsed polymer structures of the DN-DA hydrogel are very compact and rigid, and the solvent molecules that are bound to/in them are similarly limited in their mobility. Furthermore, it is clear from Table 1 that both water and acetone molecules show a similar behavior. Both solvent molecules occur either in the free state, without any restriction in its mobility, or in the bound state, where they are bound in relatively immobilized globular structures, thus indicating that the decisive factor in this behavior is, in both cases, a polar character of these molecules and hydrogen bonding. A similar behavior as that described above for water and acetone molecules from ^1^H relaxation measurements was previously found for water and ethanol molecules in poly(vinyl methyl ether)/D_2_O/ethanol solutions [48].

### 2.7. Deswelling and Swelling Kinetics

Time-dependent deswelling and swelling kinetics are shown in Figure 9 and Figure 10, respectively. To compare the effect of various stimuli, the DN-DA hydrogel samples swollen at equilibrium in water of 25 °C were immersed in pure acetone or 6 M NaCl solution or water of 45 °C. The deswelling process was monitored gravimetrically as a function of the time of deswelling (Figure 9a). After attaining the equilibrium collapsed state, the hydrogels were again immersed in water of 25 °C and the reswelling behavior was monitored until the new equilibrium state was obtained (Figure 10a). Similar procedures were performed for the water–acetone mixtures with various acetone contents and the relevant deswelling and reswelling curves are shown in Figure 9b and Figure 10b, respectively.

Swelling/deswelling processes in hydrogels are assumed to follow first-order kinetics [49,50] and the time dependence of the swelling ratio *SR*(*t*) after stimuli change could be described by the equation [51]
(1)$$SR\left(t\right)=SR_{\infty }+A_{1}\exp\left(-\frac{t}{\tau_{1}}\right)+A_{2}\exp\left(-\frac{t}{\tau_{2}}\right)$$
where *SR*_∞_ is equilibrium swelling ratio, *A*_1_ and *A*_2_ are pre-exponential factors, and *τ*_1_ and *τ*_2_ are characteristic time parameters. The measured time dependences in Figure 9 and Figure 10 showed a two-step character and it was necessary to fit them with the function with two characteristic times.

Table 2 contains the fitting characteristic time parameters *τ*_1D_, *τ*_2D_ for the deswelling curves in Figure 9 and *τ*_1S_, *τ*_2SD_ for the swelling curves in Figure 10. The swelling and deswelling curves for 20 vol.% acetone solution could not be fitted with sufficient accuracy to Equation (1) and the parameters are not included in Table 2. From Table 2, it follows that the time-dependent processes occur in two steps: a rapid process with a characteristic time *τ*_1_ = 5–15 min is followed by a slow deswelling/reswelling step, which is characterized by a time *τ*_2_ = 70–130 min. The dependences for deswelling in 80 vol.% and 100 vol.% acetone solutions showed only one characteristic time, apparently because the first process was so fast that it could not be detected. The two-step behavior of the deswelling curve is probably caused by the formation of two-phase structure, where at the beginning of the deswelling process, the outer part of the hydrogel sample deswells relatively quickly as solvent molecules have short diffusion distances for transport outside of the hydrogel. The surface part of the hydrogel thus shrinks and forms a hydrophobic barrier for the transport of solvent molecules from the inner part of the hydrogel, causing a very slow release of water and thus the longer characteristic time represented in the second step. Two-step behavior of deswelling curve was previously found for semi-interpenetrating hydrogels based on PNIPAAm [52].

Similarly, the two-step reswelling profile of the hydrogel can be explained by the formation of a two-phase structure. At the beginning of reswelling, the sample surface is in contact with water molecules, so the shrunk chains on the surface of the hydrogel begin to relax and water is easily absorbed. As the hydrogel swells, a two-structure is formed: the surface part with solvated network chains and the inner part with unswollen network chains. As the surface area of the inner part of the hydrogel decreases with increasing swelling ratio, the swelling process slows down after the initial rapid swelling. Similar two-step reswelling profile was also found and described for the PAAm hydrogels [53].

If we compare the characteristic times for the deswelling process in water–acetone mixtures, it is clear from Table 2 that, with increasing acetone content, both characteristic times decrease and the deswelling process becomes faster. In 100 vol.% acetone, the characteristic time *τ*_2D_ is practically one order of magnitude shorter compared to the deswelling time in the mixture with 30 vol.% acetone. The deswelling rate is thus obviously correlated with the change in the swelling ratio after immersion a sample from pure water into the water–acetone mixture. The DN-DA hydrogel shows the greatest change in the swelling ratio in 80 vol.% and 100 vol.% acetone (Figure 4b) and, at the same time, it has the fastest deswelling process in these mixtures compared to other water–acetone mixtures. Analogously, the same behavior can be observed when comparing the effect of various stimuli on the deswelling rate in the DN-DA hydrogel. As the DN-DA hydrogel has a relatively high swelling ratio in the 6 M NaCl solution or in pure water at 45 °C, the hydrogel deswells slowly under the influence of temperature or salt. Interestingly, the value of the time *τ*_1D_ for deswelling in 45 °C water is very low, indicating a rapid initial process of deswelling. This could be related to the faster diffusion of water molecules at a higher temperature and their ability to reach the outer part of the hydrogel quickly.

The values of characteristic times for reswelling processes in Table 2 indicate that reswelling is somewhat slower compared to deswelling. The difference is more significant for the processes that had a fast deswelling in water–acetone mixtures with 60–100 vol.% of acetone. A slower swelling in hydrogels was observed earlier, e.g., in polyacrylamide hydrogels in aqueous NaCl solutions [54] or in temperature-sensitive PNIPAAm hydrogels [50]. The dissolution of the collapsed structures can be slowed down by the existing entanglements [55]. The solvent must thus disrupt the compact globular structures, and this is a slower process than when the solvent is expelled out of the expanded hydrogel chains, which are quickly dehydrated.

Similar to deswelling, the reswelling processes are described by two characteristic times, *τ*_1S_ and *τ*_2S_. The short time *τ*_1S_ corresponding to initial stage of swelling is practically the same for all solutions in which the hydrogel was immersed, with the exception of water at temperature 45 °C, when the time *τ*_1S_ is very short due to the faster diffusion of water molecules. For water–acetone mixtures, the time *τ*_2S_ decreases with increasing acetone content, but its value is higher compared to the time *τ*_2D_ for the deswelling process.

## 3. Conclusions

The influence of various stimuli on the behavior of hydrogels with a single- and double-network structure composed of PDEAAm and PAAm was investigated. Swelling measurements and NMR spectroscopy were combined to provide information about changes in hydrogels on macroscopic and molecular scales.

The sensitivity to stimuli of the DN hydrogel was found to be significantly affected by the introduction of the second component and the formation of the double network. The interpenetrating structure in the DN hydrogel causes the units of the component, which is insensitive to the given stimulus in the form of the SN hydrogel, to be partially formed as globular structures in the DN hydrogel. The SN-D hydrogel composed of PDEAAm shows distinct temperature and salt responsiveness. Due to the hydrophilic PAAm groups, temperature- and salt-induced changes in the deswelling of the DN-DA hydrogel are less intensive and gradual compared to the SN-D hydrogel. On the other hand, the swelling ratio of the DN-DA hydrogel shows a significant decrease in dependence on the acetone content in acetone–water mixtures. This is certainly caused mainly by the presence of acetone-sensitive PAAm units, but as it was shown from the NMR experiments, 50% of acetone-insensitive PDEAAm units also contribute to the collapsed structures, which may affect the deswelling extent of the DN-DA hydrogel in water–acetone mixtures.

A certain portion of water (HDO) and acetone molecules bound in globular structures was established from the measurements of the ^1^H NMR spin–spin relaxation times *T*_2_ for the studied DN hydrogel. In water–acetone mixtures with a high content of acetone, the spin–spin relaxation times *T*_2_ for bound solvent molecules are up to three orders of magnitude lower than the relaxation time of the free molecules, signifying that the collapsed polymer structures of the DN hydrogel are very compact and rigid, and the solvent molecules that are bound to/in them are similarly limited in their mobility.

The time-dependent deswelling and reswelling kinetics showed a two-step profile, corresponding to the solvent molecules being released and absorbed during two processes with different characteristic times. The response to the given stimuli was found to be a key factor for the kinetics and the fastest deswelling and swelling processes were detected for water–acetone mixtures with a higher acetone content.

## 4. Materials and Methods

### 4.1. Swelling Experiments

#### 4.1.1. Thermo-Responsive Properties

Specimens cut having dimensions ca. 1.5 × 1.5 × 0.1 cm^3^ were swollen to reach equilibrium at room temperature (ca 25 °C) in bottles with ca. 50 mL distilled deionized water. Then, the bottles were transferred to a thermo-stated bath and equilibrated for 2 h at the starting temperature (20 °C). Masses of swollen samples were detected with a precise balance. Then, the bottles containing hydrogel samples and water were heated to a next temperature and equilibrated again for 2 h. Measurements at progressively increasing temperatures were carried out in the same way.

#### 4.1.2. Solvent- and Salt-Responsive Properties

A pre-weighed dry hydrogel sample (1.5 × 1.5 × 0.1 cm^3^) was immersed in 50 mL of the solvent for 24 h to reach equilibrium swelling, wiped out the surface water and weighed.

#### 4.1.3. Deswelling and Swelling Kinetics

The samples swollen at equilibrium in water of 25 °C were immersed in pure acetone or 6 M NaCl solution or water of 45 °C. The deswelling processes were monitored gravimetrically as a function of the time of deswelling. After attaining the equilibrium collapsed state, the samples were immersed in water of 25 °C and the reswelling behavior was monitored until the new equilibrium state was obtained.

The swelling ratio *SR* of swollen hydrogels was determined according to the formula:(2)$$SR=\frac{W_{S}-W_{D}}{W_{D}}$$
where *W_S_* and *W_D_* represent the weight of swollen and dry hydrogel, respectively.

### 4.2. ^1^H NMR Spectroscopy

Liquid-state ^1^H NMR measurements were performed with a Bruker Avance 500 liquid-state spectrometer (Bruker, Karlsruhe, Germany) operating at 500.1 MHz. The typical conditions were as follows: π/2 pulse width of 12.5 µs, relaxation delay of 20 s, spectral width of 5 kHz, acquisition time of 1.64 s and 16 scans. Sodium 2,2-dimethyl-2-silapentane-5-sulfonate (DSS) was used as an internal NMR standard. The integrated intensities were determined by spectrometer integration software with an accuracy of ±1%. Constant temperature within ±0.2 K was maintained in all measurements using a BVT 3000 temperature unit. The samples were always kept at the experimental temperature for 15 min before the measurement. To quantitatively characterize the phase transition, we used the values of the collapsed *p*-fraction (degree of collapsing) obtained as
(3)$$p=1-\frac{I}{I_{0}}$$
where *I* is the integrated intensity of the given polymer signal in the spectrum of partly collapsed hydrogel and *I*_0_ is the integrated intensity of this signal if no collapse transition occurs.

The ^1^H spin–spin relaxation times *T*_2_ of the hydrogen–deuterium oxide (HDO) and acetone were measured using the Carr–Purcell–Meiboom–Gill (CPMG) [56] pulse sequence 90°_x_-(*t*_d_-180°_y_-*t*_d_)_n_-acquisition with *t*_d_ = 0.5 ms, relaxation delay 100 s and 8 scans. The total time for the *T*_2_ relaxation was an array of 14 values.

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
