# Peer review of "External Stimuli-Responsive Characteristics of Poly(N,N′-diethylacrylamide) Hydrogels: Effect of Double Network Structure"

_gels, 2022, doi:10.3390/gels8090586_

Round 1

Reviewer 1 Report

This research work describes swelling and NMR experiments in stimuli-responsive hydrogels with single- and double-network (DN) structure composed of poly(N,N`-diethylacrylamide) and polyacrylamide (PAAm). However, some crucial NMR aspects need to be addressed:

1) Representative chemical structures of the hydrogels need to be incorporated to Figure 2 for the assignment of 1H-NMR signals.

2) The NMR spectra are not well referenced. The signal of HDO appears at 4.79 ppm. The experiments are not high-resolution, the term high-resolution is associated to High-Resolution Magic-Angle Spinning (HRMAS) NMR for swelled samples. Otherwise, it was not explained in the manuscript.

3) The NMR results are not clear. NMR experiments were done in gel phase but without using an HRMAS NMR probe. For this reason, if an NMR spectrometer was used in solution simulating the gel phase, why does the signal disappear when the mobility increases if now the hydrogels became like a liquid? More information and a rigorous studies are necessary. Also, the experimental conditions must be controlled by changing the temperature to guarantee the optimal acquisition of the NMR signals.

Reviewer 2 Report

In this paper, authors investigated SN and DN hydrogels composed of PDEAAm and PAAm with regard to sensitivity to various stimuli.  The experimental method is appropriate and the result analysis is accurate. I suggest it can be accepted in the present form.

Reviewer 3 Report

The authors are requested to put a reference concerning the particular stimulus: For example; pH (Polymers 2017, 9(4), 137); Solvent (J. Mater. Chem. A, 2021,9, 9706-9718); light (Gels 2022, 8(9), 533), ultrasound (ACS Appl. Bio Mater. 2022, 5, 7, 3212–3218); ionic strength (Biomater. Sci., 2018,6, 2073-2083) and so on…

Line 41: The collapse of hydopgels… to be changed to… The collapse of hydrogels

Please rephrase the sentence: Cononsolvency behavior was shown, for example, for PNIPAAm hydrogels in various water-organic solvents mixtures [22,23].

Please add a schematic illustration.

The authors are requested to enhance the quality of the presentation (Figures 1, 2, and 6). I’m sorry to say the presentation quality is quite poor. Please increase the font size of the X- and Y-axis.

Figure: Please correct the degree centigrade.

The authors are requested to add a comparison table between the submitted work and previous works on DN hydrogel with respective parameters.

Please take care of superscripts and subscripts throughout the text.

Reference: The author should follow the journal guidelines. Please rearrange according to the journal criteria.

Round 2

Reviewer 1 Report

The authors have been corrected and improved the previous work and answered all the concerns, so now it can be accepted.

Reviewer 3 Report

The author has addressed all suggested changes correctly. I consider the manuscript suitable for being published in gels in its present form.